

# Mentalizer
## Artificial Intelligence-Enhanced Brain-Computer Interface Framework for Real-Time User-Responsive Experience

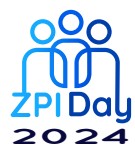

**Autors**: Paweł Dzikiewicz 0009-0003-6345-4974 · Tomasz Koralewski 0009-0008-2783-7333 · Jakub Ner 0009-0003-0936-4249 · Adam Pawłowski 0009-0000-1128-772X

**Supervisor:** Michał Kędziora

### Abstract

Brain-computer interface (BCI) technology holds immense promise for real-time adaptive systems. However, current tools often fail to effectively integrate cognitive and emotional responses into practical applications. To address this, we developed *Mentalizer* - modular BCI framework that processes EEG signals and integrates seamlessly with Unity-based applications. *Mentalizer* includes independent modules for device interfacing, pre-processing, classification, and application integration, all configurable via Unity's Scriptable Objects. This design ensures flexibility and adaptability to diverse use cases. The framework classifies cognitive states such as boredom, flow, and frustration, enabling dynamic adjustments in applications. As part of the project, we conducted experiments to gather data to train classifiers and validate the framework in practical scenarios. Use cases include dynamically adjusting video game difficulty to maintain player engagement and detecting attention lapses in high-stakes professions like airport security. *Mentalizer* demonstrates potential to enhance productivity, user satisfaction, and engagement across various domains.

## 1 INTRODUCTION

### 1.1 Background and Motivation

A brain-computer interface (BCI) is a groundbreaking technology that bridges the gap between human cognition and external devices by receiving, analyzing, and transforming brain-generated signals into actionable output commands. This innovative field enables users to control software, hardware, or other systems directly through their brain activity, bypassing conventional physical interfaces such as keyboards, mice, or touchscreens. BCIs have the potential to revolutionize a wide range of domains, including healthcare, assistive technologies, gaming, and human augmentation. By translating complex neural activity into meaningful interactions, BCIs open new possibilities for individuals with disabilities, as well as for enhancing the user experience in immersive environments.

Although initial research on BCIs began as early as the 1980s [1], it is only recently that the technology has gained traction in the consumer market and is rapidly evolving with increasing investments and new solutions [2].

Our work aims to simplify the integration of such systems, making them more accessible to everyday developers who may not have expertise in this field. Currently, solutions on the market don't integrate with each other well, which often poses a barrier to their practical implementation.

*Mentalizer* addresses this challenge by creating a toolset for the Unity environment that leverages brain-computer interfaces (BCI) and artificial intelligence to dynamically adjust the difficulty of a video game based on the user's concentration level, as inferred from brainwave activity (EEG). The project also incorporates classifiers trained on data collected during experiments conducted with the approval of an ethics committee. The idea for the project originated from our first exposure to advanced BCI equipment for EEG measurements at a neuroinformatics hackathon held in Vienna. Shortly after this event, we began planning its implementation as part of the activities of the Neuron Science Club at Wrocław University of Science and Technology and later realized it as a project within the ZPI framework.

### 1.2 Objectives

This project addresses three distinct engineering and research challenges. The first challenge involves utilizing BCI devices to gather information about the user's brainwave activity. The second challenge is the analysis and classification of this data using artificial intelligence to determine the user's states such

as boredom, engagement, frustration. The third challenge is the practical implementation of prototype applications within the Unity environment to evaluate the framework through real-world examples.

The primary business goals and added value of the project lie in creating a system capable of responding to real-time changes in user state. We have demonstrated the framework's capabilities through two examples, including its potential to maintain focus during repetitive and critical tasks such as airport baggage screening or air traffic control. Additionally, we showcased its adaptability through a video game that adjusts its difficulty in real time to match the user's focus and engagement.

## 2 LITERATURE REVIEW

When discussing mental states and emotions, it is crucial to consider the role of brainwaves—neurological reactions occurring within the brain that are closely linked to psychological states [3]. Various models have been proposed to classify and describe emotional states, including the 2D valence-arousal model [4], the 3D valence-arousal-dominance model [5], and the model proposed by Csikszentmihalyi. The latter was selected for our project as it effectively encapsulates the states we aim to assess: boredom, flow, and anxiety [6].

After identifying the emotional states of interest, our next step involved defining these states and selecting suitable preprocessing methods and classification models. Our approach was informed by prior research addressing similar problems, as well as our own experimental trials to optimize results. One foundational study that influenced our approach was *The EEG-Based BCI Emotion Recognition: A Survey 2015–2020* [7], which compared numerous classification methods and features in the context of EEG emotion recognition. Another significant reference was *Application of Artificial Intelligence Techniques for Brain–Computer Interface in Mental Fatigue Detection: A Systematic Review 2011–2022* [8]. These works emphasized the utility of frequency domain-based features for classification—a finding consistent across other studies, including *Neural Correlates of Flow, Boredom, and Anxiety in Gaming: An Electroencephalogram Study* [9] which focused on finding connection between the described emotional states and EEG signal.

For classifier selection, we analyzed methods proposed in prior research, including key studies such as *Classification of EEG Signals of User States in Gaming Using Machine Learning* [10], *Machine Learning Approaches for Boredom Classification Using EEG* [11], and *Mental Flow Estimation Through Wearable EEG* [12]. These studies provided valuable insights into the effective use of machine learning models for emotion classification, which were integrated into our framework.

Although emotional state assessment was essential for our framework, it was also necessary to analyze comparable software solutions that allow for data acquisition, information processing, AI model integration, and application connectivity. The market offers several solutions with functionalities similar to our framework, yet each has notable limitations that we aim to address. Below, we analyze three major solutions: EmotivPRO, Neurosky Developer Tools, and the Unicorn Suite by g.tec.

EmotivPRO [13] is one of the leading solutions on the market, enabling data acquisition from Emotiv devices and providing pre-trained AI algorithms for detecting mental commands. Users can leverage these features to identify intentions or actions based on EEG signals. However, this solution has notable drawbacks. The ability to create custom experiments is restricted by a subscription model, and the built-in AI algorithms are predefined and trained by the company, preventing users from modifying them or using custom models. Overall, EmotivPRO is a closed system that limits flexibility and development opportunities for developers.

Neurosky Developer Tools [14] offer a more open approach by providing access to portions of its code. However, it also comes with limitations. There is no support for integrating custom AI models, forcing users to rely on preconfigured and undocumented emotion classifiers. Furthermore, the functionality primarily focuses on data acquisition (both raw and preprocessed) and the use of built-in models. Although Neurosky provides slightly more flexibility, it still lacks the full extensibility needed for implementing advanced, custom solutions.

Unicorn Suite by g.tec [15] is another example of closed software. In this case, the primary disadvantages are the lack of capability for developers to extend its functionality. Applications integrated into the Unicorn Suite are individually priced, resulting in additional costs. Some applications feature built-in AI models, while others only allow access to preprocessed data or integrations with programming languages (e.g., Python, C). As with the previous tools, Unicorn Suite does not allow for full customization or the creation of bespoke solutions.

Finally, we assessed literature regarding affective design, which focuses on leveraging users' affective states to improve experiences. Notable examples include *Affective Level Design for a Role-Playing Videogame Evaluated by a Brain–Computer Interface and Machine Learning Method* [16], which analyzed player experiences during varied gameplay scenarios, and *A BCI-Based Assessment of a Player's State of*

*Mind for Game Adaptation* [17], which examined emotional states during a horror game. These studies demonstrated how effective design can maximize user engagement by dynamically adapting to their emotional states.

Passive monitoring of users and their activities is a popular and widely applied strategy. Many existing applications and systems implement this approach in various ways.

Eye-tracking systems for airplane pilots enable real-time monitoring of concentration, fatigue, and situational awareness. These systems facilitate the early detection of potential problems and support decision-making in critical flight moments [18].

Dynamic difficulty adjustment in video games is a well-established field that focuses on maximizing player engagement [19]. Popular strategies include adapting difficulty levels based on user performance, such as scores or progression data. More advanced but less common methods involve physiological measurements, such as heart rate, to modify gameplay elements [19].

Existing solutions for monitoring and adapting user experiences primarily focus on evaluating user performance metrics, such as response accuracy, game scores, or reaction times. While effective in many applications, these approaches face significant limitations. The most notable limitation is the inability or difficulty in accounting for the user's actual emotional state. This is largely due to technological constraints that fail to provide the necessary data for such assessments. As a result, applications and systems may overlook critical factors such as stress, frustration, or motivation. This can reduce the effectiveness of personalization and, in some cases, lead to unintended outcomes, such as user discouragement or overload.

*Mentalizer*, as a passive BCI framework for Unity, addresses these issues while retaining the advantages of standard approaches. It integrates seamlessly with existing methods, providing developers the freedom to create applications that adapt dynamically to specific needs.

# 3 ANALYSIS AND RESULTS

## 3.1 Our framework functionalities

The framework is designed in a modular structure, divided into independent components: Device, Pre-processing, Classifier, and Application. Each module operates autonomously, with data exchange facilitated through a central data bus. Additionally, the framework we proposed allows end-users to fully configure each module using Scriptable Objects - Specific to Unity data containers, enabling easy adjustment of all necessary parameters to meet the specific requirements and use premade schemes to use in the application. Figure 6 illustrates the framework's architecture in detail, highlighting the central data bus as the core of data exchange between modules. It depicts the flow of EEG data from the device through the Device Interface, followed by processing in the Data Processing module, further analysis using AI Models in the Classifier module, and integration with the Application Interface. This modular structure ensures seamless interaction with the target application, emphasizing the roles of individual components and their interconnectivity.

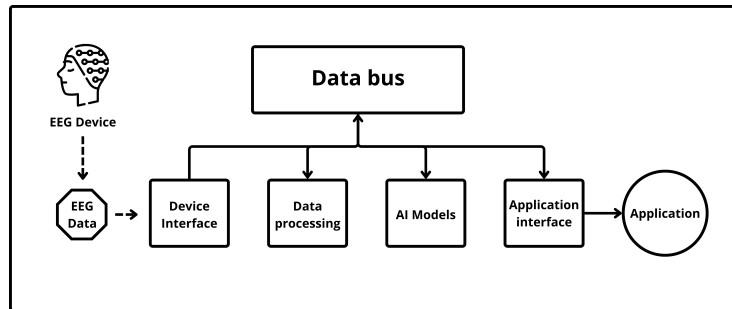

Figure 1: *Mentalizer* modules diagram. Source: own

The **Device module** is depicted as the central component for interfacing with Brain-Computer Interface (BCI) devices in figure 2. By abstracting hardware-specific details, our module provides developers with flexibility to adapt to various BCI devices and data transmission protocols. The built-in UDPStreamer is shown as a key feature, supporting UDP-based data streams and tailored for compatibility with devices like the Unicorn Hybrid Black.

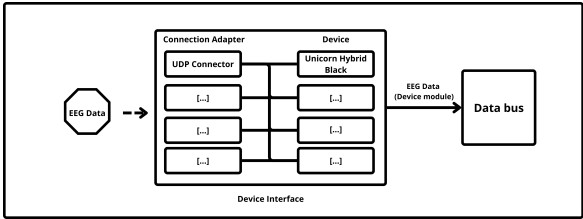

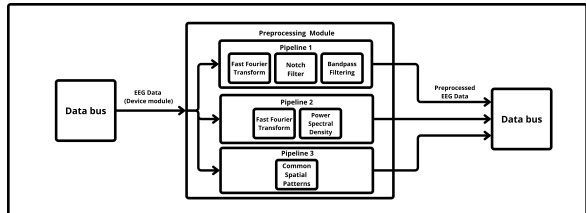

Figure 2: Device Module. Source: own

Figure 3: Preprocessing Module. Source: own

The **Preprocessing module** we implemented is the foundational stage of the data handling workflow, designed to prepare raw data for classification within the Classifier module. It is responsible for filtering, cleaning, and transforming data to ensure quality and compatibility with downstream processing. The module integrates *PythonIncluded* and *PythonMNE*, enabling precise operations on EEG data, such as frequency filtering and noise reduction. It is configured with Scriptable Objects and supports our custom pipeline architecture - figure 3.

Our **Classifier module** is responsible for data classification, utilizing previously configured models in *ONNX* format. It enables the creation and management of pipelines that define classification processes, as shown in figure 4.

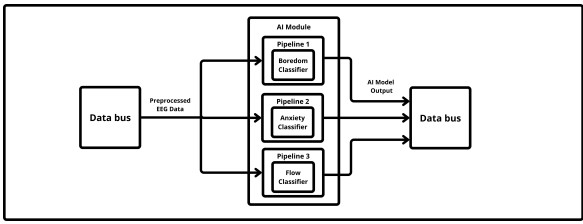

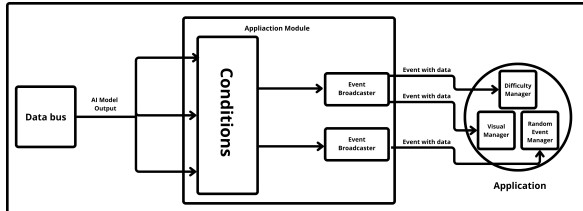

Figure 4: Classifier Module. Source: own

Figure 5: Application Module. Source: own

The **Application Module** is responsible for utilizing classified information by linking it to appropriate actions within the target application. It enables the integration of classification results with the functionality of the end system, as shown in figure 5.

The modular approach applied in our project not only ensures the independent operation of individual components but also significantly facilitates the implementation of changes and extensions to the entire mechanism. This architecture enables flexibility and scalability, adapting the system to evolving requirements and new applications.

## 3.2   Classificator creation

We developed classifiers capable of recognizing boredom, engagement, frustration, and a neutral state. To achieve this, we needed to simulate conditions that would allow us to collect and appropriately label the data. For this purpose, we obtained ethics committee approvals and designed BCI experiments using well-known computer games.

1. **Tetris:** Using Unity, we created an application consisting of three versions of Tetris, interspersed with breaks and questionnaires regarding participants' feelings during the experiment. Each version of the game was intended to elicit one of three states: boredom, flow, or a neutral state. However, the classifiers intended to distinguish these states did not perform accurately. Therefore, we limited the scope to distinguishing between two states based on this data: boredom-flow and boredom-neutral. The experiment involved 18 participants.

2. **League of Legends:** In Python, we developed Mind-Collector—a tool for gathering EEG data and enabling automatic annotation based on changes in the player's screen. Using this tool, we assessed the player's performance in real-time—identifying moments when their character died or defeated opponents. This approach allowed us to collect high-quality data, which was used to train classifiers capable of recognizing frustration and engagement. The experiment involved four participants, who collectively played 20 matches.

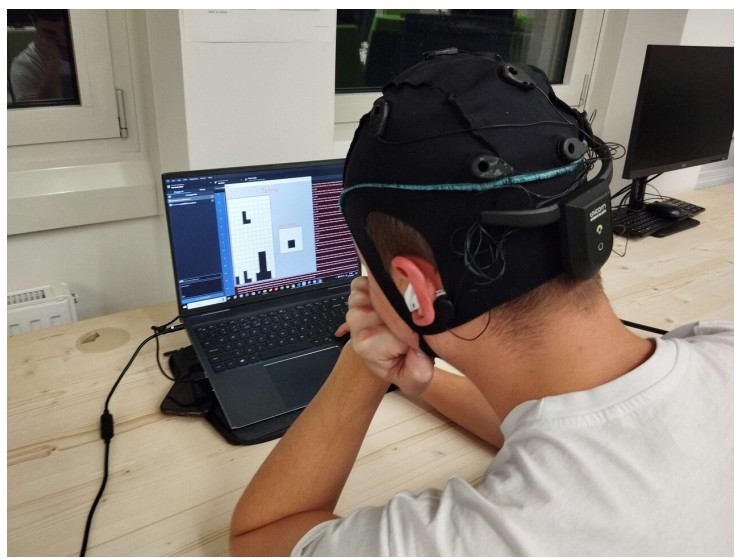

Figure 6: Participant engaging in a BCI experiment while playing Tetris. Source: own

The gathered data was transformed with Fast-Fourier Transform (FFT) from raw EEG Signal and filtered with a second order Butterworth bandpass filter ranging from 0.5 to 50 Hz. We have acquired data for frequency ranges: delta (1-4 Hz), theta (4-8 Hz), beta low (12-16 Hz), beta mid (16-20 Hz), beta high (20-30 Hz) and gamma (30-50 Hz), for each out of eight channels - Fz, C3, Cz, C4, Pz, PO7, Oz and PO8 according to the 10/20 system.

Initially, we based our models on support vector machines, which have quadratic computational complexity during fitting. However, shallow MLP neural networks proved much faster to train while achieving comparable effectiveness.

|  | Predicted: boredom | Predicted: flow |
|---|---|---|
| **Actual: boredom** | 6047 | 1413 |
| **Actual: flow** | 1177 | 6331 |

*Confusion Matrix for a Multilayer Perceptron (MLP) Classifying States of Boredom and Immersion.*

|  | precision | recall | f1-score | support |
|---|---|---|---|---|
| **boredom** | 0.837 | 0.811 | 0.824 | 7460.0 |
| **flow** | 0.818 | 0.843 | 0.830 | 7508.0 |
| **accuracy** | 0.827 | 0.827 | 0.827 | 0.827 |

*Metrics Report for a Multilayer Perceptron (MLP) Classifying States of Boredom and Immersion.*

### 3.3   Use cases

EEG technologies enable the monitoring of users' cognitive states, unlocking new possibilities across various fields – from workplaces where concentration is critical, to games that require dynamic gameplay personalization. As part of our project, we developed two use cases: the first involves the work of X-ray scanner operators at airports, and the second focuses on dynamically adjusting the difficulty level in a survival game. In both cases, we analyze EEG data to enhance user efficiency and engagement.

### Analysis of Boredom in X-ray Scanner Operators at Airports

X-ray scanner operators at airports often perform repetitive tasks, which can lead to boredom and a decline in concentration. Our developed system monitors the operator's cognitive state using EEG data, focusing on detecting boredom. When the system detects signs of boredom, a simulated image of a potentially dangerous object, such as a weapon or blade, is displayed on the screen.

This stimulus aims to capture the operator's attention and restore focus on the image being analyzed. This mechanism helps reduce the risk of overlooking actual threats, thereby enhancing safety and work efficiency.

**Maintaining the *Flow* State and Combatting Boredom in a Survival Game**

In survival games, it is crucial to maintain the player in a *flow* state, which is the balance between challenge and engagement. Our system analyzes EEG data, focusing on two states: boredom and engagement. Based on this, it dynamically manages the pace of the gameplay by changing the frequency of enemy appearances.

- **Example 1: Signs of Boredom** – When the system detects low player engagement, it increases the frequency of enemy appearances, adding dynamism and raising the level of challenge.

- **Example 2: Maintaining *Flow*** – The system continuously adjusts the game pace to provide optimal challenges, preventing a decline in motivation or frustration for the player.

Thanks to this approach, the game becomes more immersive and engaging, and the dynamic difficulty adjustment allows for maintaining the player's interest for a longer period as per the rules of affective level design. [16]

# 4 DISCUSSION

## 4.1 Practical implications

The *Mentalizer* framework enables a wide range of applications in work, education, and entertainment, significantly enhancing user experiences. Drawing on Csikszentmihalyi's Flow Model [6], it helps maintain users in a *flow* state, fostering full engagement and optimal performance.

A major strength of *Mentalizer* is its real-time analysis of emotional states, such as boredom, frustration, and engagement, allowing dynamic adaptation to user needs. Unlike rigid algorithms, our system offers superior flexibility and personalization, ensuring a balanced and engaging experience by keeping users in their *flow* zone. This adaptability extends from incremental adjustments to impactful stimuli, such as simulated threats for baggage controllers, improving efficiency in monotonous yet critical tasks.

The framework also enhances education by enabling more responsive and engaging learning experiences. Cognitive state analysis helps tailor curricula and teaching styles to better meet student needs, improving educational outcomes. In workplace applications, it supports monitoring employee concentration, optimizing schedules, planning breaks, and identifying periods of decreased focus to better align tasks and meetings with productivity peaks.

*Mentalizer* stands out for its versatility, supporting integration with various EEG devices, custom algorithms, and AI models, while allowing extensive personalization in end-user applications. Its modular structure ensures adaptability across fields—from video games to education and industrial decision-support systems.

Though our examples focus on three cognitive states, *Mentalizer* is a universal tool for creating tailored applications, making it an invaluable resource for advancing brain-computer interface technologies.

## 4.2 Challenges and Limitations

*Mentalizer*, despite its numerous advantages, also has some limitations. Currently, it does not support simultaneous operation with multiple devices at once. Additionally, the framework is tightly integrated with the Unity system. On the one hand, this allows for easy use of features such as scriptable objects, making it accessible even for beginner developers. However, this approach restricts its use to applications within the Unity ecosystem

## 4.3 Future work

In the future, significant improvements and expansions are planned for *Mentalizer*. First and foremost, support for multiple devices will be added, enabling more advanced applications. A module supporting transfer learning is also planned, which will allow for faster adaptation of models to new users and scenarios on the go. The database of supported EEG devices, signal preprocessing effects, and classifiers will be expanded, making the framework more versatile. Additionally, further scientific studies are planned to support the development of the technology and document its effectiveness in various applications.

# 5 CONCLUSIONS

## 5.1 Conclusions

The development of *Mentalizer* demonstrates the potential of brain-computer interfaces (BCIs) to revolutionize real-time user interaction by integrating cognitive state analysis into Unity-based applications. Through a modular design and adaptability, we have created the that framework bridges the gap between EEG signal acquisition, processing, and real-world applications, offering a versatile toolset for developers and researchers alike. Key contributions of this work include:

- **Innovation in Cognitive State Analysis**: By leveraging artificial intelligence, *Mentalizer* successfully classifies cognitive states such as boredom, engagement, and frustration. These classifiers were developed using EEG data collected from 18 participants in controlled experiments, providing a robust foundation for dynamic adaptations in applications.

- **Real-World Applications**: Use cases such as dynamic difficulty adjustment in video games and attention restoration in airport security highlight the practical value of our approach. These solutions were implemented and validated within the framework, addressing real-world challenges in diverse fields, including entertainment, education, and workplace productivity.

- **Modular and Extensible Framework**: The framework's architecture allows for independent module operation, ensuring scalability and adaptability to evolving technological and application-specific needs. Scriptable Objects in Unity enable users to configure modules effortlessly and integrate custom classifiers or preprocessing algorithms.

- **Scientific and Technical Achievements**: The project integrates rigorous data collection, classifier development, and practical implementation, underpinned by ethical research practices. Performance metrics demonstrate the reliability and precision of cognitive state identification using EEG data.

The results of our tests confirm the effectiveness of the *Mentalizer* framework in dynamically adjusting applications based on EEG signals to enhance user experience. In games and in simulated airport security monitoring, the system successfully identified cognitive states like boredom and engagement, adapting the gameplay or providing focus-restoring stimuli. This dynamic adjustment improved user engagement and performance, demonstrating the positive impact of real-time cognitive state integration on interactive experiences. The framework shows potential for enhancing user satisfaction across various fields.

## 5.2 Acknowledgements

We acknowledge that this study involved research with human participants and was conducted in full compliance with ethical standards. Approval was obtained from the "Komisji ds. Etyki Badań Naukowych Politechniki Wrocławskiej" prior to the initiation of the study, and all participants provided informed consent in accordance with the guidelines outlined by the committee.

We would like to sincerely thank everyone who has supported us. In particular, our fellow students from our department, as well as other volunteers, for their assistance in collecting the data necessary for training the models. We would also like to thank Michele Romani for his valuable support and help in resolving technical challenges.

An additional source of motivation for us was the recognition our project received during the global Neurohackathon *NeurotechX* 2023, the AI Innovators 2024 conference in Warsaw, the finals of the "Młode Talenty 2024" competition, and the Polish final of "Red Bull Basement 2024". These accolades served as proof that the concept of our project is intriguing, and that our work holds potential for further development.

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
