# OpenReview forum: "Artificial Intelligence-Enhanced Brain-Computer Interface Framework for Real-Time User-Responsive Experience"
_pwr.edu.pl/Wrocław_University_of_Science_and_Technology/2024/ZPI_Day — Wrocław University of Science and Technology 2024 ZPI Day Submission_

### Official Review · Reviewer_jdKy · 2024-12-03
**Artificial Intelligence-Enhanced Brain-Computer Interface Framework for Real-Time User-Responsive Experience**

**Confidence:** 5
**Significance Of Results:** 5
**Overall Quality:** 5

**Compliance With Template:**

5: Very High Quality – The article contains all the required sections, which are written in a very detailed, clear, and error-free manner. The structure is professional and meets expectations, and the content adheres to the highest substantive and formal standards.

**Description Of Results:**

5: Very High Quality – The results are described in detail, clearly and comprehensively, supported by thorough evaluation, analysis, and convincing usage examples. The description meets the highest substantive standards.

**Feedback On Consistency:**

Consistency of the project description is very high quality.  Problem analysis, presentation of results, and conclusions are consistent and logical.

**Potential For Development:**

Yes, article not only indicate possibilities for further work or practical applications of its results. Project itself inslude practical application of the results. Project will be continued within student research group.

**Project Nature Evaluation:**

Project not only exhibit characteristics of an engineering work, such as the level of utility, application of technical methods, and technological solutions, but also exceeds it with experimental part of project which was aligned with ethics committee and conducted within the project. Project represents implementing working framework for EEG signal acquisition, processing, and real-world applications.

**Technical Language Precision:**

5: Very High Quality – The language is entirely appropriate for a technical report. All terms are used correctly and precisely, and the style is professional, clear, and coherent, without any errors or ambiguities.

---

### Official Review · Reviewer_wBRK · 2024-12-05
**Artificial Intelligence-Enhanced Brain-Computer Interface Framework for Real-Time User-Responsive Experience**

**Confidence:** 5
**Significance Of Results:** 5
**Overall Quality:** 5

**Compliance With Template:**

5: Very High Quality – The article contains all the required sections, which are written in a very detailed, clear, and error-free manner. The structure is professional and meets expectations, and the content adheres to the highest substantive and formal standards.

**Description Of Results:**

5: Very High Quality – The results are described in detail, clearly and comprehensively, supported by thorough evaluation, analysis, and convincing usage examples. The description meets the highest substantive standards.

**Feedback On Consistency:**

The results of tests confirm the effectiveness of the Mentalizer framework in dynamically adjusting applications based on EEG signals to enhance user experience.

**Potential For Development:**

The project can be practically applied.

**Project Nature Evaluation:**

The project shows the characteristics of engineering work.

**Technical Language Precision:**

5: Very High Quality – The language is entirely appropriate for a technical report. All terms are used correctly and precisely, and the style is professional, clear, and coherent, without any errors or ambiguities.

---

### Official Review · Reviewer_Ev9x · 2024-12-06
**The review of Mentalizer**

**Confidence:** 4
**Significance Of Results:** 5
**Overall Quality:** 5

**Compliance With Template:**

5: Very High Quality – The article contains all the required sections, which are written in a very detailed, clear, and error-free manner. The structure is professional and meets expectations, and the content adheres to the highest substantive and formal standards.

**Description Of Results:**

5: Very High Quality – The results are described in detail, clearly and comprehensively, supported by thorough evaluation, analysis, and convincing usage examples. The description meets the highest substantive standards.

**Feedback On Consistency:**

Paper is clear and easy to read.

Minor errors:
 - in the Acknowledgement section: "Approval was obtained from the "Kommisji ds Etyki"...
The nominative case of polish name of the institution is required, instead of genitive.

**Potential For Development:**

Authors clearly state their future work. They show that more studies are needed to make the project more mature.

**Project Nature Evaluation:**

Mentalized is a framework for BCI interfaces  supplemented with AI, it helps with implementation of real-time adjustment of applications with the EEG data. The project has combined nature, both scientific and utlitary. Authors describe the technical and scienfitic aspects of adopted methods as well as selected applications.

**Technical Language Precision:**

4: High Quality – The language is appropriate for a technical report. Terminology is used correctly, and statements are precise, with only minor shortcomings that do not affect the overall clarity.

---

### Decision · Program_Chairs · 2024-12-10

Accept (Oral)